# Developing a taxonomy to describe offspring outcomes in studies involving pregnant mammals' exposure to non-tobacco nicotine: A systematic scoping review

Lucy Phillips[1]*, Ross Thomson[1], Tom Coleman-Haynes[1], Sue Cooper[1], Felix Naughton[2], Lisa Mcdaid[2], Joanne Emery[2], Tim Coleman[1]

1 Centre for Academic Primary Care, University of Nottingham, Nottingham, United Kingdom, 2 School of Health Sciences, University of East Anglia, Norwich, United Kingdom

* l.phillips1@nottingham.ac.uk

## Abstract

### Introduction

Many countries recommend Nicotine Replacement Therapy (NRT) for smoking cessation in pregnancy. Preclinical studies of nicotine exposure to pregnant mammals could indicate how nicotine may adversely affect the developing fetus. As a first step towards summarising this literature, we undertook a systematic scoping review to determine the number and nature of offspring outcomes studied.

### Methods

We searched MEDLINE and EMBASE databases for papers reporting empirical data on offspring outcomes following nicotine exposure to pregnant non-human mammals. We excluded studies that investigated exposure to only smoking, e-cigarettes, nicotine vaccines, or studies with no 'nicotine only' group. We developed a draft taxonomy and using this, described and quantified outcomes reported.

### Results

We identified 476 studies, which reported 729 offspring outcomes. The draft taxonomy classified outcomes as being measured in i) whole animals, ii) body systems and iii) 'other'. Body system outcomes were further categorised as being functional changes, or changes at macroscopic or cellular levels. The most frequently used outcomes were those detecting changes in the brain (n = 265), physical parameters measured in whole animals (n = 122) and any respiratory system changes (n = 97).

### Conclusions

This scoping review quantifies the nature and frequency of outcomes used in preclinical studies investigating the potential impacts of nicotine administration in pregnancy on offspring. Systematic reviews of studies investigating outcomes involving animal brains,

**Data Availability Statement:** All relevant data are within the paper and its Supporting Information files.

**Funding:** This project was funded through a National Institute for Health Research (NIHR) Programme Grants for Applied Research (Programme number RP-PG-0615-20003). Tim Coleman is a National Institute for Health Research (NIHR) Senior Investigator. The views expressed in this manuscript are those of the author(s) and not necessarily those of the NIHR or the Department of Health and Social Care.

**Competing interests:** The authors have declared that no competing interests exist.

respiratory system, or 'whole animal' outcomes may have greatest potential for further advancing knowledge regarding impacts of gestational nicotine exposure on offspring.

## Protocol and registration

Protocol for this review can be found on Open Science Framework (https://osf.io/ptmzc/).

## Introduction

Smoking in pregnancy causes much morbidity and mortality [1–3]. A systematic review of randomised control trials (RCTs) shows that stopping smoking in pregnancy improves birth outcomes [4]. Many countries, including the UK, recommend nicotine replacement therapy (NRT) for smoking cessation during pregnancy [5, 6] and 11% of UK pregnant smokers receive NRT prescriptions [7]. Although NRT provides nicotine without other toxic elements present in tobacco smoke, the ability for nicotine to cross the placenta and concentrate in fetal blood and amniotic fluid [8] leads to concerns that nicotine within NRT may cause fetal harm [9].

Preclinical literature could inform us of potential harms that might be due to nicotine, and many preclinical studies have investigated relationships between nicotine administered to pregnant animals and adverse outcomes [10]. However, there is no agreed core outcome set for measuring potential offspring harm from nicotine and there appears to be great diversity in outcomes used. For many preclinical study outcomes, there is no direct evidence that these are indicative of harm to humans; hence, such outcomes are probably best considered theoretically indicative of harm to infants born to pregnant women who use NRT. For example, in one study fetal brain stem injury was observed in rat offspring after gestational nicotine administration in maternal drinking water, and it was extrapolated that maternal smoking in pregnancy may cause similar issues, potentially leading to sudden infant death syndrome [11]. Other preclinical study issues which make finding's relevance to humans difficult to ascertain include outcomes measured 'in vitro', in isolated parts of animals or in species which have little in common with humans [12].

Systematic reviews involve collating all studies relevant to research questions together, and through evidence synthesis can comprehensively answer important research questions[13]. For example, a recent systematic review of studies investigating NRT use by pregnant women found no evidence that NRT use during pregnancy causes fetal harm [14]. However, despite a thorough search, we have been unable to find any systematic reviews of preclinical studies investigating the impacts of nicotine exposure in pregnancy on mammalian offspring outcomes. We found a 20-year old review, but within this, search strategies were not described, so it was impossible to determine how comprehensive this was [10]. Other reviews have been narrative with selective choices of included literature [8, 15–18], and this lack of systematic preclinical literature synthesis has resulted in calls for rigorous, regularly-updated systematic reviews of preclinical studies to focus research efforts [19].

Scoping reviews allow rapid identification of unanswered research questions, and may indicate both the feasibility and potential utility of conducting full systematic reviews [20]. We have conducted a systematic scoping review of preclinical studies investigating potentially adverse impacts of gestational nicotine administration on offspring outcomes. We identify, quantify and categorise the full range of offspring outcomes measured in studies involved, and then use the taxonomy developed to hypothesise about the utility of preclinical literature for

identifying potential adverse effects of nicotine on human offspring following administration of NRT in pregnancy.

## Methods

We adapted the framework for conducting scoping reviews described by Arskey and O'Malley [20]. This involved: posing a research question; identifying potentially relevant studies; selecting those appropriate for inclusion; extracting and charting relevant data; and then collating, summarizing and reporting findings. We used established methods [21] to comprehensively search for preclinical studies in which non-tobacco nicotine has been administered to mammals in pregnancy, and we identified those which reported fetal or offspring outcomes.

### Search strategy (identifying relevant studies)

We developed a draft MEDLINE search strategy using keywords relevant to the research question and tailored terms to make the search results more specific. We then tested the draft search strategy in MEDLINE to confirm it could find three pre-identified relevant papers, before this was finalised and adapted to run in both MEDLINE and Embase. In MEDLINE, we used the option to search for non-human studies only. However, as this option was not available for EMBASE, we applied a search filter which had been specially developed to identify all animal studies [22]. No past date restrictions were applied to either search, and these were conducted by 21st September 2022. A full list of final search terms used in study searches is outlined in S1 File.

We downloaded all identified citations into Endnote, removed duplicates and, where only titles were present, sought abstracts where possible. We then exported all citations into Covidence [23] online systematic review software, which facilitated management of the screening process. The search strategy captured journal articles in any language that reported empirical data on fetal or offspring outcomes following the administration of non-tobacco nicotine to pregnant animals. Due to the broad nature of the animal/non-human filters applied, some non-mammalian animal studies were present in search results. These were excluded during the initial title screening. Foreign language titles and abstracts were translated; studies with a title but no abstract were included if a relevant outcome was clearly stated. We decided on inclusion or exclusion using study titles and abstracts only. Although there is evidence that research papers can show inconsistencies between their abstract and full text [24], this is rarely an issue for identifying outcomes or the population of investigation.

### Screening strategy (Study selection)

Initially, LP screened citations (i.e. titles +/- abstracts) and those that were obviously not relevant to the review were excluded. Next, using pre-determined inclusion and exclusion criteria, three reviewers (LP, RT and TC) each piloted study selection procedures on 100 citation records. Individual assessors noted uncertainties and discussed these with the other two researchers. Following resolution of issues arising from the pilot screening, and based on increasing familiarity with the literature, inclusion and exclusion criteria for the scoping review were finalised (Table 1). LP and RT then applied the inclusion and exclusion criteria to all the citations. Citations were grouped into those for inclusion, exclusion or where a decision could not be made. Next, all reviewers (LP, RT & TC) discussed citations where there was uncertainty about whether to include or exclude them.

### Data extraction (charting the data)

A list of citations with an abstract were divided between two researchers (LP & TCH) who independently extracted the following: author name, publication date, animal species used and

**Table 1. Inclusion and exclusion criteria.**

| Inclusion Criteria | Exclusion Criteria |
|---|---|
| • Published journal papers (not conference abstracts or reviews)<br>• Papers reporting empirical data on fetal or offspring outcomes following nicotine exposure to pregnant mammals<br>• Papers in any language (foreign language abstracts will be translated) | • Studies that do not include pregnant mammals or fetal/ offspring nicotine exposure during gestation (e.g in vitro studies)<br>• Studies which investigate exposure to only smoking, e-cigarettes, nicotine vaccines, studies with no 'nicotine only' comparison group<br>• Studies that include nicotine exposure to offspring in the postpartum (e.g via lactation) prior to outcome measurement. |

all reported outcomes (no information on nicotine exposure e.g. amount, duration or administration was extracted). For citations with no abstract, researchers extracted any outcome data that was mentioned in the title. If the title did not state a clear outcome, and no abstract was available, the citation record was excluded from the scoping review. A spreadsheet listing all outcomes extracted from abstracts and other information outlined above was compiled. As some papers used technical, discipline-specific language, where necessary, we included in the spreadsheet a lay translation of the outcomes that would assist in the draft taxonomy development. Once all outcomes had been extracted, a third researcher (RT) conducted an audit of 10% of the data extraction and lay translations to ensure consistency between researchers. Many citation records reported more than one outcome so the denominator for findings is the number of reported outcomes rather than the number of citation records.

## Taxonomy development (Collating, summarizing and reporting the results)

We adopted an iterative approach to developing an outcomes taxonomy and initially grouped outcomes as either physical (e.g. weight, litter size, survival) or relating to specific body systems (e.g. brain, cardiovascular or respiratory systems). After reflecting, we judged some 'body system' categories too broad to be sufficiently informative and so divided these further to better describe the diversity of reported outcomes. Other categories judged to be too narrow were merged; for example, we created an overarching 'Nervous System' category by combining the categories for outcomes relating to the Central, Autonomic and Peripheral Nervous Systems. We also added an 'other' category for any outcomes that did not fit into either the 'physical' or 'body system' categories (for example, blood gasses), and this category also included any outcome parameters that were unclear. We proposed and revised category descriptors to try to make these as consistent as possible with other preclinical literature. For example, a draft descriptor 'psychological outcomes' was thought too 'human' and replaced with the descriptor 'brain function', which encompassed all cognitive or behavioural outcomes.

## Results

Searches produced 6354 citation records, 1798 of which were duplicates. Following title and abstract screening of the remaining 4556 records, 476 records were assessed as relevant for inclusion in the scoping review (Fig 1). We were unable to obtain abstracts for seven of the 476 citation records, meaning outcomes for these citations were determined by the title alone.

We identified 729 fetal and offspring outcomes within the 476 abstracts or titles. The majority of papers (440) reported outcomes relating to rodents, while 16 related to sheep, 14 to primates and 3 to rabbits. The draft taxonomy for categorising outcomes contains three

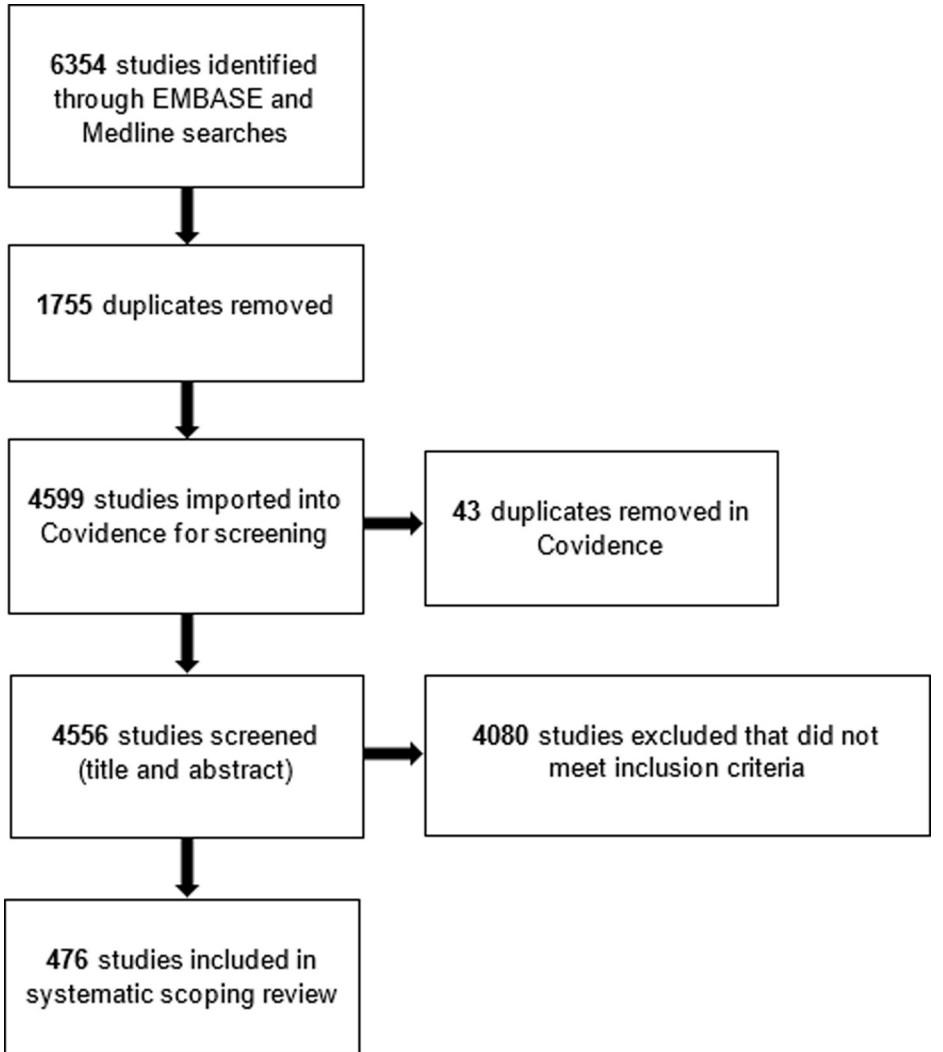

**Fig 1. PRISMA diagram.** Prisma diagram of citation selection process.

classification groups: outcomes measured in i) complete animals, ii) body systems and, iii) 'other' (see Table 2).

## 1. Outcomes measured in complete animals

This comprised any outcomes measured in whole animals, such as litter size, survival, weight or length. Measures were taken either at birth or soon afterwards, and animals were alive at the time of measurement or killed immediately beforehand. In this category, 122 outcomes were distributed across 91 studies; most were weight or length (87/122).

## 2. Outcomes measured in body systems

This comprised outcomes that could be measured in complete animals but were more relevant to a physiological body system (e.g. cardiovascular, neurological or renal systems). Within each body system, further subcategories defined outcomes as cellular-level (e.g. hormones, DNA or chemical changes), macroscopic (i.e. directly observable without microscopic-level

**Table 2. Draft taxonomy for categorising outcomes in preclinical literature.**

| Type of Outcome | Outcome Subcategory | 2nd Level subcategory | 2nd Level Subcategory totals | Subcategory Total | Outcome Total |
|---|---|---|---|---|---|
| **Whole animal** | Litter Size | - | - | 14 | 122 |
| | Survival | - | - | 21 | |
| | Weight & Length | - | - | 87 | |
| **Body System** | Brain | Cellular (hormones, DNA, chemical changes) | 174 | 265 | 593 |
| | | Macroscopic development | 20 | | |
| | | Function | 73 | | |
| | Cardiovascular | Cellular (hormones, DNA, chemical changes) | 26 | 65 | |
| | | Macroscopic development | 8 | | |
| | | Function | 31 | | |
| | Eyes | Cellular (hormones, DNA, chemical changes) | 2 | 4 | |
| | | Macroscopic development | 2 | | |
| | | Function | - | | |
| | Gastrointestinal system (liver, pancreas, etc.) | Cellular (hormones, DNA, chemical changes) | 17 | 20 | |
| | | Macroscopic development | 2 | | |
| | | Function | 1 | | |
| | Immune system/ immune function | Cellular (hormones, DNA, chemical changes) | 12 | 15 | |
| | | Macroscopic development | 1 | | |
| | | Function | 2 | | |
| | Musculoskeletal, skin & dental | Cellular (hormones, DNA, chemical changes) | 17 | 36 | |
| | | Macroscopic development | 16 | | |
| | | Function | 3 | | |
| | Nervous System (CNS, ANS, PNS, etc.) | Cellular (hormones, DNA, chemical changes) | 14 | 21 | |
| | | Macroscopic development | - | | |
| | | Function | 7 | | |
| | Neuroendocrine & endocrine System | Cellular (hormones, DNA, chemical changes) | 28 | 32 | |
| | | Macroscopic development | 1 | | |
| | | Function | 3 | | |
| | Renal | Cellular (hormones, DNA, chemical changes) | 13 | 18 | |
| | | Macroscopic development | 4 | | |
| | | Function | 1 | | |
| | Reproductive system | Cellular (hormones, DNA, chemical changes) | 11 | 18 | |
| | | Macroscopic development | 2 | | |
| | | Function | 5 | | |
| | Respiratory | Cellular (hormones, DNA, chemical changes) | 45 | 97 | |
| | | Macroscopic development | 14 | | |
| | | Function | 38 | | |
| **Other** | Fetal Blood Gases | - | - | 7 | 14 |
| | Intrauterine Development/ Developmental Parameters | - | - | 7 | |

identification), or functional (affecting a body system, e.g. lung function measures). Of 593 outcomes, 45% (265/593) related to the brain, 16% (97/593) to respiratory, 11% (65/593) to cardiovascular, 6% (36/593) to musculoskeletal, skin and dental, and 5% (32/593) to neuroendocrine or endocrine systems. Outcomes relating to other body systems had 21 or fewer occurrences. Across all body systems, 359 outcomes were cellular-level, 70 were macroscopic and 164 related to body system function.

### 3. Other outcomes

This encompassed any outcomes that could not be classified into another category. For example, fetal arterial blood gas concentrations were a reported outcome in some studies, but these can occur due to disturbances in a number of body systems so categorisation is difficult. There were 14 outcomes that fell into this category.

## Discussion

We have provided a thorough, quantitative description of outcomes used in preclinical studies investigating mammalian prenatal nicotine exposure. We identified 729 different fetal or offspring outcomes reported from 476 studies. The most frequently reported outcomes measured cellular-level brain changes; respiratory system outcomes measuring either cellular, macroscopic or functional changes, and physical outcomes measured in whole animals, such as birth weight and litter size.

Many studies reported multiple outcomes and, as we only extracted data from titles and abstracts; some outcomes reported only in manuscripts' full texts may have been missed. However, authors usually include the most important study outcomes in abstracts, so we should have identified those outcomes which researchers most strongly believed to be related to maternal nicotine administration. Also, as authors are likely to be biased towards including statistically significant findings in abstracts, we have probably also identified most outcomes for which statistical associations with nicotine administration were demonstrated. If we had retrieved and extracted information from manuscripts' text, we may have found extra methodological details to prompt either exclusion from or inclusion in the review. For example, some studies might have involved nicotine administration in both pregnancy and the weaning period, without study abstracts mentioning this. Hence, some included studies may have assessed impacts other than those of nicotine administration in pregnancy. It is unclear how much these factors might have affected the number and breadth of outcomes identified, but there is no reason to suspect effects would be substantial.

A strength of this work is its novelty. It is the first to offer a systematic and comprehensive overview of principal offspring outcomes in animal studies investigating the impacts of nicotine in pregnancy, and the draft taxonomy is the first attempt to categorise these. Another study strength lies with its rigour. We followed established methods for conducting scoping reviews and carefully developed search strategies for identifying relevant work. This included, for one of the bibliographic databases searched, applying a filter designed to find all preclinical studies. Hence, we believe that we will have missed few relevant papers. Similarly, we used a robust, team-orientated approach to extract data; two researchers independently conducted this with arbitration from a third, where necessary. Consequently, we believe that most outcomes have been identified from citations and that these will have been categorised consistently. To ensure that the draft taxonomy was appropriately descriptive, we adopted an iterative developmental approach, revising categories and descriptors to help the final version have the greatest possible utility. However, this draft taxonomy can be revised if necessary.

This work helps move the literature beyond narrative reviews [15–19]. Although the manuscript makes no quality assessments, by demonstrating which outcomes are most regularly reported, it highlights where further investigation might be most fruitful. Detailed systematic reviews would most likely provide useful information if these answered focussed questions about impacts on the brain, respiratory system or on whole animals' physical characteristics. Such reviews could investigate the effects of varying nicotine doses and administration regimens on the same outcomes across all preclinical studies, and by including robust evaluation included studies' methodological quality, would enable effective assessment of their findings' validity. Such reviews would very likely help us better understand any relationships between gestational nicotine exposures and offspring harm in preclinical studies.

Whilst a systematic approach to synthesising the preclinical literature could indicate 'signals' of potential harms from nicotine, the question of how findings relate to humans and in particular, the impacts of NRT used in pregnancy for smoking cessation remain difficult to answer. Whether or not nicotine administered in pregnancy is harmful to human infants can only be fully answered by studies conducted in women.

## Conclusions

We provide an overview of preclinical, mammalian study outcomes used to investigate impacts on offspring from prenatal nicotine exposure, and a draft taxonomy that categorises outcomes in relation to offspring's development, affected body systems and their functions. This highlights those areas where most preclinical research has focused, and therefore could benefit most from full systematic reviews to fully understand any relationships between gestational nicotine administration and offspring harms.

## Supporting information

**S1 File. Search term for database searches.** Search strategy used to search MEDLINE and Embase.
(PDF)

**S2 File. Included citations dataset.** Information for citations included in the systematic scoping review.
(PDF)

**S3 File. Preferred Reporting Items for Systematic reviews and Meta-Analyses extension for Scoping Reviews (PRISMA-ScR) checklist.**
(PDF)

## Author Contributions

**Conceptualization:** Sue Cooper, Felix Naughton, Tim Coleman.

**Formal analysis:** Lucy Phillips, Ross Thomson, Tom Coleman-Haynes, Tim Coleman.

**Funding acquisition:** Tim Coleman.

**Investigation:** Lucy Phillips, Ross Thomson, Tim Coleman.

**Methodology:** Lucy Phillips, Ross Thomson, Tom Coleman-Haynes, Sue Cooper, Felix Naughton, Lisa Mcdaid, Joanne Emery, Tim Coleman.

**Writing – original draft:** Lucy Phillips, Ross Thomson, Tim Coleman.

**Writing – review & editing:** Lucy Phillips, Ross Thomson, Sue Cooper, Felix Naughton, Lisa Mcdaid, Joanne Emery, Tim Coleman.

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
