## [Decision Letter · Decision Letter 0]

26 Aug 2022

PONE-D-22-05338Developing a taxonomy to describe offspring outcomes in studies involving pregnant mammals’ exposure to non-tobacco nicotine: a systematic scoping reviewPLOS ONE

Dear Dr. Phillips,

Thank you for submitting your manuscript to PLOS ONE. After careful consideration, we feel that it has merit but does not fully meet PLOS ONE’s publication criteria as it currently stands. Therefore, we invite you to submit a revised version of the manuscript that addresses the points raised during the review process. While nearly ready for publication, we request that you address the following-- Please ensure that your PRISMA flowchart details the reasons for record exclusion at each stage, and;- Systematic and scoping reviews should provide an up-to-date overview of the research topic covered. Since the literature search was completed over a year ago, we also request that your update your search to be more current. 

We look forward to receiving your revised manuscript.

Kind regards,

Avanti Dey, PhD

Staff Editor

PLOS ONE

Journal Requirements:

3. Please include captions for your Supporting Information files at the end of your manuscript, and update any in-text citations to match accordingly. Please see our Supporting Information guidelines for more information: 

Reviewers' comments:

Reviewer's Responses to Questions

**Comments to the Author**

1. Is the manuscript technically sound, and do the data support the conclusions?

Reviewer #1: Yes

2. Has the statistical analysis been performed appropriately and rigorously? 

Reviewer #1: N/A

3. Have the authors made all data underlying the findings in their manuscript fully available?

Reviewer #1: Yes

4. Is the manuscript presented in an intelligible fashion and written in standard English?

Reviewer #1: Yes

5. Review Comments to the Author

Reviewer #1: This is a useful contribution to the literature surrounding the use of NRT during pregnancy. This scoping review is well laid out and has identified areas where further research could be conducted. I have no suggestions to improve the clarity of this manuscript

6. PLOS authors have the option to publish the peer review history of their article (what does this mean?). If published, this will include your full peer review and any attached files.

Reviewer #1: No

---

## [Author Response · Author response to Decision Letter 0]

6 Oct 2022

We have made the following edits to the manuscript and supporting information as requested. The PRISMA diagram now includes human or non-mammalian participants, conference abstracts or reviews, and exposure to nicotine via lactation as reasons for exclusion. The searches were completed again on 21st September 2022 and any new relevant citations have been added to the final taxonomy figures. The manuscript has been updated to reflect the guidance provided in your email. File names have also been amended to follow these requirements. We have also now added a supporting information file that includes the title, author and publication year for all citations included in the systematic scoping review. The references included in the manuscript have been checked and formatting amended where needed.

---

## [Decision Letter · Decision Letter 1]

10 Jan 2023

Developing a taxonomy to describe offspring outcomes in studies involving pregnant mammals’ exposure to non-tobacco nicotine: a systematic scoping review

PONE-D-22-05338R1

Dear Dr. Phillips, 

We’re pleased to inform you that your manuscript has been judged scientifically suitable for publication and will be formally accepted for publication once it meets all outstanding technical requirements.

Kind regards,

Saeed Ahmed, MD

Academic Editor

PLOS ONE

---

## [Editor Report · Acceptance letter]

26 Jan 2023

PONE-D-22-05338R1 

Developing a taxonomy to describe offspring outcomes in studies involving pregnant mammals’ exposure to non-tobacco nicotine: a systematic scoping review 

Dear Dr. Phillips:

I'm pleased to inform you that your manuscript has been deemed suitable for publication in PLOS ONE. Congratulations! Your manuscript is now with our production department. 

Kind regards, 

on behalf of

Dr. Saeed Ahmed 

Academic Editor

PLOS ONE